# GAT-Flow: Predictor-Corrector Flow Matching with Graph Attention Network for Crystalline Materials

## Abstract

Crystalline materials discovery empowered by deep generative models is critical for driving progress in applications such as energy storage, electronics, and catalysis. However, current approaches face significant challenges in accurately predicting complex structures and ensuring specific properties, thereby hindering their practical applicability. In this work, we propose GAT-Flow, a flow-based generative framework designed to address these challenges. We leverage a graph attention network to jointly predict lattice vectors and atomic coordinates, effectively capturing both local coordination and periodic patterns. We also incorporate a Predictor-Corrector sample strategy to improve sampling efficiency and numerical stability. In addition, by leveraging training-free guidance from a pre-trained language model, we enable property-driven crystalline generation based on textual prompts. Experimental results demonstrate that GAT-Flow achieves state-of-the-art performance in crystalline structure prediction. Moreover, our approach enables material generation with specific properties, offering new perspectives on structure-property alignment in computational materials design.

## 1 Introduction

Crystalline materials discovery with tailored properties has driven breakthroughs in energy storage (Chen et al., 2022b), electronics (Schweidler et al., 2024), and catalysis (Chen et al., 2022a). Predicting stable atomic configurations from vast structural spaces—termed Crystal Structure Prediction (CSP) and De Novo Generation (DNG)—remains a fundamental challenge in computational materials design. These tasks are inherently NP-hard (Stillinger, 1999): as the number of atoms per unit cell increases linearly, the potential energy surface grows exponentially, creating an astronomical number of local minima. Traditional approaches relying on Density Functional Theory (DFT) (Kohn & Sham, 1965) face scalability limitations due to their computational cost, making exhaustive exploration infeasible for systems beyond modest sizes (Pickard & Needs, 2011; Yamashita et al., 2018b; Wang et al., 2010b; Zhang et al., 2017).

To address the computational challenges associated with traditional methods, recent developments in generative modeling have proposed alternative frameworks, including those based on variational autoencoders, diffusion models, and flow-based models. Notable examples include DiffCSP (Jiao et al., 2023), CrystalFlow (Luo et al., 2024), MatterGen (Zeni et al., 2025), etc. Despite these advances, several limitations persist. First, these methods do not incorporate Graph Attention Network (GAT), which limit their ability to capture long-range interactions, leading to physically implausible outputs. Second, the iterative sampling procedures employed in these models accumulate numerical errors and are computationally expensive. Finally, most existing frameworks lack inherent property-driven generation capabilities, thus necessitating costly post-processing and DFT calculations (Zeni et al., 2025; Sriram et al., 2024) to generate materials with desired properties.

In this work, we propose GAT-Flow, a continuous normalizing flows (CNFs) model which improves the accuracy of structure prediction and enables efficient structure generation. In the context of the CSP task, our method employs fractional coordinates to represent crystal structures, thereby encoding their periodicity. Within a Conditional Flow Matching (CFM) framework, we jointly predict the lattice vectors and fractional coordinates using a geometric GAT, with the prediction further refined

through a Predictor-Corrector (PC) sampling strategy. Our method integrates two key innovations: geometric graph attention network and an efficient Predictor-Corrector sampling strategy. These components are specifically designed to directly address the aforementioned limitations.

Through comprehensive experiments on real-world datasets and synthetic benchmarks, we demonstrate that GAT-Flow outperforms state-of-the-art generative models in both reconstruction accuracy and sampling efficiency within the context of CSP. Furthermore, our LLM-integrated DNG strategy achieves strong alignment between generated structures and target properties, highlighting the potential of incorporating external knowledge sources into generative frameworks for materials discovery.

In summary, our main contributions are as follows:

**Geometric Graph Attention Network**    We improve equivariant graph neural networks (Satorras et al., 2021) by proposing a geometric multi-layer GAT model. Within the network, we first apply symmetry transformations to the edge features and node features of crystal structure to satisfy equivariance with respect to the crystal symmetries. Subsequently, multi-head attention mechanism is employed to process the transformed edge and node features, capturing the interaction relationships within the crystal. Finally, residual connections are used to integrate the outputs from each network layer, generating the target structure.

**Efficient Predictor-Corrector Sampling Strategy**    To address the computational inefficiency and numerical instability of high-order Ordinary Differential Equation (ODE) solvers in flow-based models, we introduce a PC sampling framework inspired by Langevin dynamics. This method first utilizes an ODE solver for trajectory prediction, followed by the introduction of a stochastic perturbation term derived from the Langevin equation to explore the local space around the predicted trajectory. This combination of deterministic and stochastic elements improves prediction accuracy while reducing computational costs.

## 2 RELATED WORK

In traditional computational materials science, researchers have predominantly relied on Density Functional Theory (DFT) to identify energetically stable crystal structures. While optimization algorithms such as Bayesian optimization (Yamashita et al., 2018a), genetic algorithms (Yamashita et al., 2022), and particle swarm optimization (Wang et al., 2010a) have been successfully employed for generating and screening candidate structures, the inherent computational complexity of DFT calculations presents a fundamental challenge in balancing efficiency with accuracy.

Recently, generative models have demonstrated significant potential in the field of crystal structure prediction. Modern approaches leverage powerful frameworks including Variational Autoencoders (VAEs), Generative Adversarial Networks (GANs), and diffusion models, demonstrating remarkable potential in this domain. Notably, architectures such as CDVAE (Xie et al., 2022) and SyMat (Luo et al., 2023) integrate VAEs with score-based diffusion models to operate directly on atomic coordinates, while maintaining Euclidean and periodic invariance through equivariant graph neural networks (GNNs). DiffCSP (Jiao et al., 2023) advances this direction by formulating structure prediction as a joint optimization of atomic coordinates and lattice parameters within a unified diffusion framework. Further pushing the boundaries of efficiency, FlowMM (Miller et al., 2024) introduces Riemannian flow matching to crystal generation tasks, achieving superior sampling performance. To enhance the capability of crystalline materials discovery, we propose the GAT network, improve the sample strategy, and integrate LLM to develop our flow-based model, GAT-Flow.

## 3 PRELIMINARIES

Our method involves modeling probability distributions over crystal lattices, which are defined as periodic arrangements of atoms in three-dimensional space. A crystal lattice is generated by periodically repeating a basic building block known as the unit cell, which contains a particular configuration of atoms. This repetition extends infinitely to form the entire crystalline structure. In this section, we provide a summary of crystal representation, laying the groundwork for the detailed description of our model in Section 4. Additional background on crystal can be found in Appendix A.

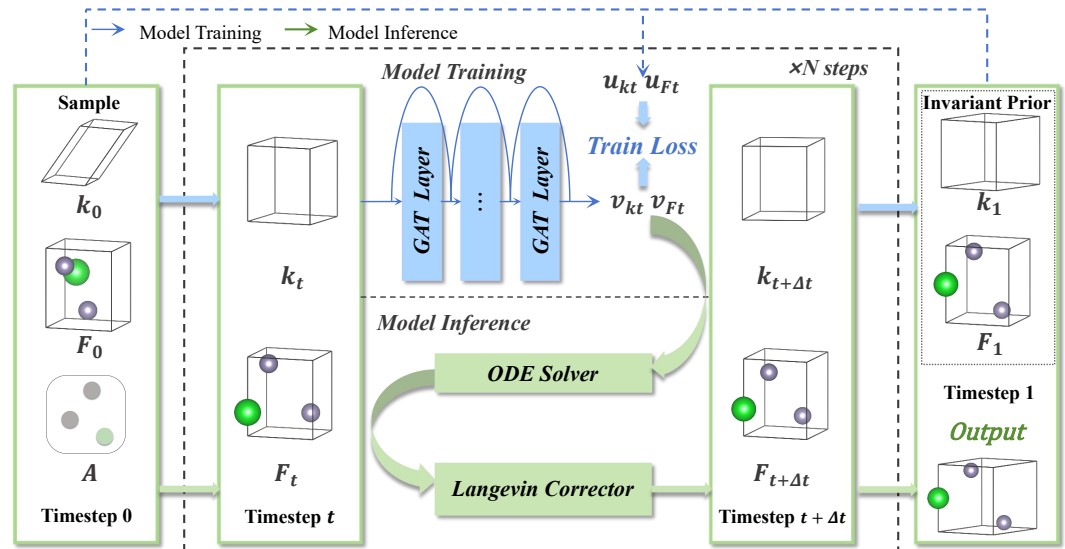

Figure 1: GAT-Flow Model Architecture. Given the chemical composition $A$, we denote $\mathbf{k}_t$ and $\mathbf{F}_t$ as the lattice matrix and fractional coordinate matrix at time $t$, respectively. Vector fields $u_t^k$ and $u_t^F$ define the continuous normalizing flows at time $t$. The GAT gets $u_t^k, u_t^F$ as inputs and outputs $v_t^k, v_t^F$.

In this paper, we represent a crystal containing $n \in \mathbb{N}$ atoms as a triple:

$$\mathbf{C} := (\mathbf{A}, \mathbf{F}, \mathbf{L}) \in \mathcal{C},$$

where $\mathbf{A} \in \mathcal{A}^n$ denotes atom types, $\mathbf{F} \in \mathcal{F} = [0,1)^{n \times 3}$ represents fractional coordinates of atomic positions, and $\mathbf{L} \in \mathcal{L} \subset \mathbb{R}^{3 \times 3}$ encodes the geometry of the unit cell. The atomic type matrix $\mathbf{A} = [a^1, \ldots, a^n]^\top$ represents the type of the $i$-th atom, with each $a^i \in \mathcal{A}$. The position matrix $\mathbf{F} = [f^1, \ldots, f^n]^\top$ records the coordinates of each atom, where each $f^i \in [0,1)^3$ is expressed in fractional coordinates under periodic boundary conditions. Finally, the lattice matrix $\mathbf{L}$ encapsulates all geometric information of the unit cell, wherein $a, b, c > 0$ denote the edge lengths of the unit cell in Angstroms (Å), and $\alpha, \beta, \gamma \in [60°, 120°]$ represent the internal angles between the edges.

Our crystal representation exhibits several symmetries induced by the following group actions: permutation invariance, translation invariance, and rotational invariance (Kondor & Trivedi, 2018). The geometric graph attention network we employed inherently incorporates permutation invariance. To effectively model the rotational invariance of lattice parameters, we adopt a parameterization inspired by DiffCSP++ Jiao et al. (2024), which decouples rotational and structural components of the unit cell. Specifically, the lattice matrix $\mathbf{L}$ is expressed through a polar decomposition:

$$\mathbf{L} = Q \exp\left(\sum_{i=1}^{6} k_i B_i\right),$$

where $Q$ is an orthogonal matrix representing rotational degrees of freedom and $\{B_i\}_{i=1}^6$ forms an orthonormal basis for the space of symmetric $3 \times 3$ matrices, and $\mathbf{k} = (k_1, \ldots, k_6)^\top \in \mathbb{R}^6$ encodes the shape of the lattice in a rotation-invariant manner. This rotational invariance of lattice and the fractional coordinate system can fulfill the translation invariance with respect to $\mathbf{F}$ (Luo et al., 2024).

## 4 METHOD

Upon the specification of the external conditions $\mathbf{c}$, CSP task predicts the lattice matrix $\mathbf{k}$ and the fractional matrix $\mathbf{F}$ given its chemical composition $\mathbf{A}$ as condition for each unit cell while DNG task predicts $\mathbf{k}, \mathbf{F}, \mathbf{A}$ (detailed in B). We propose GAT-Flow, a graph attention-based continuous normalizing flow model tailored for CSP task. Within the framework of conditional flow matching, we employ a graph attention network to model and predict both lattice matrix and fractional coordinates. To enhance the accuracy of CSP task, we apply Langevin-style correction steps during the sampling phase. Obviously, our framework can be readily extended to DNG task. The overall architecture of the proposed GAT-Flow model is illustrated in Fig. 1.

### 4.1 JOINT FLOW MODELING

Our model takes $(\mathbf{k}_0, \mathbf{F}_0)$ as inputs and produces $(\mathbf{k}_1, \mathbf{F}_1)$ as outputs. To model the complex data distribution of crystal structures, we utilize CNFs Chen et al. (2018) based on a smooth, time-dependent vector field $u : [0, 1] \times \mathbb{R}^d \to \mathbb{R}^d$, defined by an ordinary differential equation (ODE):

$$\frac{d\mathbf{k}}{dt} = u_t^k(\mathbf{k} \mid \mathbf{c}), \quad \frac{d\mathbf{F}}{dt} = u_t^F(\mathbf{F} \mid \mathbf{c}). \tag{1}$$

The ODE solution starting from initial state describes the evolution of $\mathbf{k}$ and $\mathbf{F}$ over time. By modeling $u_t^k(\mathbf{k} \mid \mathbf{c})$ and $u_t^F(\mathbf{F} \mid \mathbf{c})$ using neural networks $v_{t;\theta}(\mathbf{k}, \mathbf{c})$ and $v_{t;\theta}(\mathbf{F}, \mathbf{c})$, we can transform a simple initial density into a complex target density.

Due to the complexity of calculating the vector fields $u_t^k(\mathbf{k} \mid \mathbf{c})$ and $u_t^F(\mathbf{F} \mid \mathbf{c})$, we employ CFM Tong et al. (2023), which introduces an additional conditioning variable $z$, making $u_t^k(\mathbf{k} \mid \mathbf{c}, z)$ and $u_t^F(\mathbf{F} \mid \mathbf{c}, z)$ easier to calculate. Then, the probability paths $p_t^k(\mathbf{k} \mid \mathbf{c})$ and $p_t^F(\mathbf{F} \mid \mathbf{c})$ can be represented as a mixture of conditional probability paths $p_t^k(\mathbf{k} \mid \mathbf{c}, z)$ and $p_t^k(\mathbf{k} \mid \mathbf{c}, z)$, namely, $p_t(\cdot \mid \mathbf{c}) = \int p_t(\cdot \mid z)q(z \mid \mathbf{c})dz$. Considering the continuity equation $\partial p_t / \partial t = -\nabla \cdot (p_t u_t)$, we have

$$-\nabla \cdot (p_t(\cdot \mid \mathbf{c})u_t(\cdot \mid \mathbf{c})) = \frac{\partial p_t(\cdot \mid \mathbf{c})}{\partial t} = \int \frac{\partial p_t(\cdot \mid z)}{\partial t}q(z \mid \mathbf{c})dz$$
$$= -\nabla \cdot \left( \int p_t(\cdot \mid z)u_t(\cdot \mid z)q(z \mid \mathbf{c})dz \right). \tag{2}$$

Therefore, the vector fields $u_t^k(\mathbf{k} \mid \mathbf{c})$ and $u_t^F(\mathbf{F} \mid \mathbf{c})$, which generate the probability path $p_t^k(\mathbf{k} \mid \mathbf{c})$ and $p_t^F(\mathbf{F} \mid \mathbf{c})$, are given by:

$$u_t^k(\mathbf{k} \mid \mathbf{c}) = \int \frac{u_t^k(\mathbf{k} \mid z)p_t^k(\mathbf{k} \mid z)q(z \mid \mathbf{c})}{p_t^k(\mathbf{k} \mid \mathbf{c})}dz, \tag{3}$$

$$u_t^F(\mathbf{F} \mid \mathbf{c}) = \int \frac{u_t^F(\mathbf{F} \mid \mathbf{c}, z)p_t^F(\mathbf{F} \mid z)q(z \mid \mathbf{c})}{p_t^F(\mathbf{F} \mid \mathbf{c})}dz. \tag{4}$$

In practice, We use the GAT network output $v_{t;\theta}^k(\mathbf{k}, \mathbf{c})$ and $v_{t;\theta}^F(\mathbf{F}, \mathbf{c})$ to model $u_t^k(\mathbf{k} \mid \mathbf{c}, z)$ and $u_t^F(\mathbf{F} \mid \mathbf{c}, z)$ respectively. Further details are in Appendix C.

**Flow on Lattice Matrix** The lattice matrix is parameterized using polar decomposition parameters $\mathbf{k}$. For computational convenience, the initial state distribution $p_t^k(\mathbf{k}_0 \mid \mathbf{c}, z)$ is artificially defined as a Gaussian prior $\mathcal{N}(\mathbf{k}_0; \boldsymbol{\mu}_0^k, (\boldsymbol{\sigma}_0^k)^2\mathbf{I})$, with $\boldsymbol{\mu}_0^k = (0, 0, 0, 0, 0, 1)$ and $\boldsymbol{\sigma}_0^k = 0.1$. For the conditional probability path $p_t^k(\mathbf{k} \mid \mathbf{c}, z)$, we suppose that $\boldsymbol{\mu}_t^k = t\mathbf{k}_1 + (1 - t)\mathbf{k}_0$, and $\boldsymbol{\sigma}_t^k = \boldsymbol{\sigma}^k$. Thus, the conditional probability path and vector field are given by:

$$p_t^k(\mathbf{k} \mid \mathbf{c}, z) = \mathcal{N}\left(\mathbf{k} \mid t\mathbf{k}_1 + (1 - t)\mathbf{k}_0, (\boldsymbol{\sigma}^k)^2\mathbf{I}\right), \tag{5}$$

$$u_t^k(\mathbf{k} \mid \mathbf{c}, z) = \frac{d\boldsymbol{\mu}_t^k}{dt} = \mathbf{k}_1 - \mathbf{k}_0. \tag{6}$$

Setting $\boldsymbol{\sigma}^k = 0$ reduces the conditional path to deterministic interpolation, consistent with the Rectified Flow framework Luo et al. (2024); Liu et al. (2022).

**Flow on Fractional Coordinates** To handle the periodic nature of crystal structures, the push-forward distribution $p_t^F(\mathbf{F} \mid \mathbf{c}, z)$ must satisfy translational invariance under periodic boundary conditions. To avoid introducing any bias, We define the prior distribution over fractional coordinates $p_t^F(\mathbf{F}_0 \mid \mathbf{c}, z)$ as a uniform distribution $\mathcal{U}(\mathbf{F}_0; [0, 1]^{3 \times N})$, where $N$ is the number of atoms in the unit cell. For the conditional probability path $p_t^k(\mathbf{k} \mid \mathbf{c}, z)$, we suppose that $p_t^k(\mathbf{k} \mid \mathbf{c}, z)$ follows $\mathcal{N}_w\left(\mathbf{F}; \boldsymbol{\mu}_t^F, (\boldsymbol{\sigma}_t^F)^2\mathbf{I}\right)$, $\boldsymbol{\mu}_t^F = \mathbf{F}_0 + t\left[w(\mathbf{F}_1 - \mathbf{F}_0 - 0.5) + 0.5\right]$, and $\boldsymbol{\sigma}_t^F = \boldsymbol{\sigma}^F$, where $w(x) = x - \lfloor x \rfloor$ ensures values remain within the unit interval. Thus, the conditional probability path and vector field can be formulated as:

$$p_t^F(\mathbf{F} \mid \mathbf{c}, z) = \mathcal{N}_w\left(\mathbf{F}; \mathbf{F}_0 + t\left[w(\mathbf{F}_1 - \mathbf{F}_0 - 0.5) + 0.5\right], (\boldsymbol{\sigma}^F)^2\mathbf{I}\right), \tag{7}$$

$$u_t^F(\mathbf{F} \mid \mathbf{c}, z) = \frac{d\boldsymbol{\mu}_t^F}{dt} = w(\mathbf{F}_1 - \mathbf{F}_0 - 0.5) + 0.5. \tag{8}$$

where $\mathcal{N}_w$ denotes the wrapped Gaussian distribution which ensures the periodicity of the probability distribution at each time $t$:

$$\mathcal{N}_w(\mathbf{F}; \boldsymbol{\mu}, \boldsymbol{\sigma}^2) \propto \sum_{\mathbf{Z} \in \mathbb{Z}^{3 \times N}} \exp\left[-\frac{\|\mathbf{F} - \boldsymbol{\mu} + \mathbf{Z}\|^2}{2\boldsymbol{\sigma}^2}\right]. \tag{9}$$

By setting $\boldsymbol{\sigma}^F = 0$, we align with Rectified Flow framework Luo et al. (2024); Liu et al. (2022), yielding deterministic interpolation between initial and final atomic configurations. See more details in Appendix D.

Integrating flows over both lattice parameters and fractional coordinates enables effective modeling of crystal structures while preserving physical constraints like periodicity and geometric plausibility. This approach enhances the expressiveness of the generative model and ensures sampled structures adhere to fundamental material symmetries.

## 4.2 GEOMETRIC GRAPH ATTENTION NETWORK

We design a geometric graph attention network which introduces a unified attention mechanism to dynamically allocate multi-dimensional feature weights by modeling node and edge features. We utilize a multi-layer GAT network with residual connections which enables more accurate capture of complex atomic interactions long-range dependencies in crystalline materials. Our geometric graph attention layer (GATL) takes as input the set of node features $\mathbf{h}^l = \{h_0^l, \ldots, h_{M-1}^l\}$, coordinate embeddings $\mathbf{F}^l = \{f_0^l, \ldots, f_{M-1}^l\}$, atom type embeddings $\mathbf{A} = \{a_0, \ldots, a_{M-1}\}$ and lattice embeddings $\mathbf{k}$ and outputs a transformation on $\mathbf{h}^{l+1}$ and $\mathbf{F}^{l+1}$. Concisely: $\mathbf{h}^{l+1}, \mathbf{F}^{l+1} = \text{GATL}[\mathbf{h}^l, \mathbf{x}^l, \mathbf{k}]$. We define edge features between atoms $i$ and $j$ as:

$$\mathbf{e}_{ij} = [h_i^l; h_j^l; \varphi(\Delta f_{ij}^l); \mathbf{k}], \tag{10}$$

where $\Delta f_{ij} = f_j - f_i$ is the coordinate difference, $\phi(\cdot)$ applies fourier transformer. Node feature $i$ is defined as $h_i^l = \varphi_0(f_t(t), f_A(a_i)) + \varphi_y(f_y(\mathbf{c} \setminus \mathbf{A}))$, where $\varphi_\square$ represents multi-layer perceptrons and $\mathbf{c} \setminus \mathbf{A}$ represents conditions excluding $\mathbf{A}$ (Jiao et al., 2023; 2024). Here, $f_t$, $f_A$, and $f_y$ correspond to sinusoidal positional encoding, atomic embedding, and Gaussian basis encoding, respectively. We design a multi-head attention layer with $K$ heads to handle edge features:

$$\hat{\mathbf{e}}_{ij} := BN\left(\sum_{k=1}^{K} \text{softmax}\left(f_e^k(\mathbf{e}_{ij}(\mathbf{e}_{ij})^T)\right)\right), \tag{11}$$

where $f_e(\cdot)$ denotes neural network layer and $BN(\cdot)$ denotes batch normalization. The edge features are updated as follows:

$$\mathbf{e}_i = \sum_{j \in N(i)} \Delta f_{ij}^l \mathbf{e}_{ij}, \quad \hat{\mathbf{e}}_i := \frac{1}{|\mathcal{N}(i)|} \sum_{j \in i} \hat{\mathbf{e}}_{ij}, \tag{12}$$

where $|\mathcal{N}(i)|$ denotes the set of neighbors of node $i$. The node features are updated by:

$$\hat{h}_i := f_\alpha(h_i^l, \mathbf{e}_i), \quad h_i^{l+1} = h_i^l + BN\left(\sum_{k=1}^{K} \text{softmax}\left(f_h^k(\hat{h}_i(\hat{h}_i)^T)\right)\right), \tag{13}$$

where $f_\alpha(\cdot)$ and $f_h(\cdot)$ denote neural network layers. Coordinate embeddings are then updated by: $f_i^{l+1} = f_i^l + \hat{\mathbf{e}}_i$. After message passing through $L$ GATL layers, the vector field is read out by:

$$v_{t;\theta}^k(\mathbf{k}, \mathbf{A}) = f_k\left(\frac{1}{N} \sum_{i=1}^{N} \mathbf{h}^L\right), \quad v_{t;\theta}^F(\mathbf{F}, \mathbf{A}) = f_F(\mathbf{h}^L), \tag{14}$$

where $f_k$, $f_F$ denotes multi-layer perceptrons.

Our GAT network governs the evolution of lattice and fractional coordinates within the continuous-time normalizing flow, ensuring physically valid and symmetry-preserving crystal generation.

### 4.3 MODEL TRAINING AND INFERENCE

We adopt the CFM Liu et al. (2022) framework for modeling crystal structures, with the training objective:

$$L_{\text{CFM}} = \mathbb{E}_{t,x_0,x_1} \left[ \lambda_k \left\| v_{t;\theta}^k(\mathbf{k}, \mathbf{A}) - u_t^k(\mathbf{k}|\mathbf{A}, z) \right\|^2 + \lambda_F \left\| v_{t;\theta}^F(\mathbf{F}, \mathbf{A}) - u_t^F(\mathbf{F}|\mathbf{A}, z) \right\|^2 \right], \quad (15)$$

where $x_1$ represents $(\mathbf{k}_1, \mathbf{F}_1)$, $x_0$ represents $(\mathbf{k}_0, \mathbf{F}_0)$, the vector fields $v_{t;\theta}^k(\mathbf{k}, \mathbf{A})$ and $v_{t;\theta}^F(\mathbf{F}, \mathbf{A})$ are time-dependent vector field parameterized by the GAT network with learnable parameters $\theta$. $\lambda_k, \lambda_F$ denote weighting coefficients.

This strategy ensures that the GAT network accurately learns the dynamics of the CFM framework, enabling stable and property-conditioned generation of crystal structures.

To generate physically valid crystal structures, we propose a PC sampling strategy, combining global prediction and local refinement for improved quality and stability.

Sampling proceeds in reverse time from $t = 1$ to $t = 0$ over $N$ steps with fixed step size $\Delta t = 1/N$, ensuring smooth transitions without manual scheduling. The inference process consists of:

**Predictor Step** In the predictor step, we implement two ODE solvers to update the state: the explicit Euler method and the second-order Taylor expansion method. The Euler method, a first-order approximation, updates the state as:

$$\mathbf{k}_{t-\Delta t}^{\text{pred}} = \mathbf{k}_t + \Delta t \cdot v_{t;\theta}^k(\mathbf{k}, \mathbf{A}), \quad \mathbf{F}_{t-\Delta t}^{\text{pred}} = \mathbf{F}_t + \Delta t \cdot v_{t;\theta}^F(\mathbf{F}, \mathbf{A}), \quad (16)$$

This method is computationally simple but limited in accuracy. For improved accuracy, the second-order Taylor expansion includes the first derivative term:

$$\mathbf{k}_{t-\Delta t}^{\text{pred}} = \mathbf{k}_t + \Delta t \cdot v_{t;\theta}^k(\mathbf{k}, \mathbf{A}) + \frac{\Delta t^2}{2} \cdot \frac{\partial v_{t;\theta}^k(\mathbf{k}, \mathbf{A})}{\partial t}, \quad (17)$$

$$\mathbf{F}_{t-\Delta t}^{\text{pred}} = \mathbf{F}_t + \Delta t \cdot v_{t;\theta}^F(\mathbf{F}, \mathbf{A}) + \frac{\Delta t^2}{2} \cdot \frac{\partial v_{t;\theta}^F(\mathbf{F}, \mathbf{A})}{\partial t}, \quad (18)$$

Since the external condition $\mathbf{A}$ is time-independent, the time derivative term is expanded as:

$$\frac{\partial v_{t;\theta}^k(\mathbf{k}, \mathbf{A})}{\partial t} = \frac{\partial v_{t;\theta}^k}{\partial t} + \frac{\partial v_{t;\theta}^k}{\partial \mathbf{k}} \cdot v_{t;\theta}^k(\mathbf{k}, \mathbf{A}), \quad \frac{\partial v_{t;\theta}^F(\mathbf{F}, \mathbf{A})}{\partial t} = \frac{\partial v_{t;\theta}^F}{\partial t} + \frac{\partial v_{t;\theta}^F}{\partial \mathbf{F}} \cdot v_{t;\theta}^F(\mathbf{F}, \mathbf{A}), \quad (19)$$

While the second-order method improves accuracy, it requires additional derivative calculations, increasing computational cost.

**Corrector Step** The evolution of the sample distribution $p_t(x)$ during the ODE solution follows: $\frac{\partial p_t(x)}{\partial t} = -\nabla \cdot (p_t(x)v(t, x))$, which is a deterministic flow process that does not introduce new sample distribution models and is susceptible to pattern collapse. Langevin-style correction steps are incorporated in (Karras et al., 2022) as: $dx_t = -\nabla U(x_t)dt + \sqrt{2D}dW_t$, where $U(x)$ is the potential energy function, $D$ is the diffusion coefficient, and $W_t$ denotes the Wiener process (Brownian motion). This term is superimposed on the deterministic update term $v(t, x)dt$ in the ODE, thus extending the original deterministic flow process to a flow process with noise. After correction, we get the corresponding Fokker-Planck equation (Risken & Risken, 1996) as $\frac{\partial p_t(x)}{\partial t} = -\nabla \cdot (p_t(x)v(t, x)) + \frac{\eta^2}{2}\nabla^2 p_t(x)$, where the second term represents Gaussian noise diffusion, which randomly perturbs samples within localized regions to broaden the coverage of the distribution.

Based on the above certification, we apply the following correction:

$$\mathbf{k}_{t-\Delta t}^{\text{corr}} = \mathbf{k}_{t-\Delta t}^{\text{pred}} + \eta_k \cdot \epsilon_k, \quad \mathbf{F}_{t-\Delta t}^{\text{corr}} = \mathbf{F}_{t-\Delta t}^{\text{pred}} + \eta_F \cdot \epsilon_F, \quad (20)$$

where $\epsilon \sim \mathcal{N}(0, I)$ are independent standard Gaussian noises and $\eta$ is a small noise scale (e.g., $\eta \sim 0.01$). After correction, fractional coordinates are wrapped back into the unit cell via: $F_{t-\Delta t}^{\text{corr}} \leftarrow F_{t-\Delta t}^{\text{corr}} \mod 1$, ensuring physical consistency under periodic boundary conditions. The final structure over the time interval $t \in [0, 1]$ is then reconstructed as:

$$\mathbf{k}_1 = \mathbf{k}_0 + \int_0^1 s(t)\mathbf{k}_t^{\text{corr}}dt, \quad \mathbf{F}_1 = \mathbf{F}_0 + \int_0^1 s(t)\mathbf{F}_t^{\text{corr}}dt, \quad (21)$$

Figure 2: Text-guided framework for property-driven crystalline materials generation. At each step, we perturb both global and local crystal features, use a fine-tuned LLM to predict band-gap energies, and select the closest match for the next iteration.

where $s(t) := 1 + s't$ is a scaling term part of the anti-annealing numerical scheme (Miller et al., 2024; Luo et al., 2024).

Langevin dynamics-based Corrector enhances sampling diversity by introducing stochastic perturbations that mitigate mode collapse and refine local features through simulated thermal fluctuations. This sampling strategy effectively avoids the accumulation of errors in ODE solvers.

Table 1: Results on MP-20 and MPTS-52 in CSP tasks, with the best results in bold. Performance is evaluated using Match Rate (MR) and Root Mean Square Error (RMSE).

|  | # of samples | MP-20 MR(%)↑ | RMSE↓ | MPTS-52 MR(%)↑ | RMSE↓ |
|---|---|---|---|---|---|
| CDVAE | 1 | 33.90 | 0.1045 | 5.34 | 0.2106 |
| DiffCSP | 1 | 51.49 | 0.0631 | 12.19 | 0.1786 |
| FlowMM | 1 | 61.39 | 0.0566 | 17.54 | 0.1726 |
| CrystalFlow | 1 | 59.06 | 0.1270 | 18.73 | 0.1608 |
| GAT-Flow (ours) | 1 | **62.90** | **0.0546** | **24.41** | **0.1228** |
| CDVAE | 20 | 66.95 | 0.1026 | 20.79 | 0.2085 |
| DiffCSP | 20 | 77.93 | **0.0492** | 34.02 | 0.1749 |
| CrystalFlow | 20 | 78.08 | 0.0577 | 38.55 | 0.1703 |
| GAT-Flow (ours) | 20 | **78.76** | 0.0523 | **40.96** | **0.1516** |

Table 2: Ablation studies on MP-20 using Match Rate (MR) and Root Mean Square Error (RMSE). $AttF$ and $AttL$ denotes attention block on **F** and **L**, respectively.

|  | Match rate (%) ↑ | RMSE ↓ |
|---|---|---|
| GAT-Flow | **62.90** | **0.0546** |
| **w/o GAT** | | |
| w/o $AttF$ | 62.25 | 0.0653 |
| w/o $AttL$ | 61.57 | 0.0699 |
| **w/o PC strategy** | | |
| w/ Euler solver | 60.89 | 0.071 |
| w/ Taylor solver | 60.26 | 0.065 |

## 5 EXPERIMENTS

In this section, we present the experimental setup in §5.1 and demonstrate the superior performance of GAT-Flow in CSP tasks in §5.2. Furthermore, we show the capability of GAT-Flow combined with LLM in property-driven DNG tasks in §5.3.

### 5.1 EXPERIMENTAL SETUP

**Dataset Description**    We conduct experiments on two widely recognized benchmark datasets, MP-20 and MPTS-52 (Jain et al.). MP-20 comprises 45,231 stable inorganic materials from the Material Projects (Jain et al.), mostly experimentally derived compounds with up to 20 atoms per unit cell. MPTS-52 extends MP-20 with 40,476 structures containing up to 52 atoms per cell, ordered by earliest publication year. For MP-20, we use the 60–20–20 split following (Xie et al., 2022); for MPTS-52, the split is 27,380 training, 5,000 validation, and 8,096 test samples in chronological order following (Jiao et al., 2023).

**Comparison Method**    This study focuses on performance of generative models. We compare our model to four prior methods, including variational autoencoders (CDVAE), conditional diffusion

model (DiffCSP), flow-matching-based models (FlowMM, CrystalFlow). For previously published results, we cite the findings from accepted papers; for models that have not yet been published but are available as open-source, we report our own replication results.

**Evaluation Metrics** For CSP task, we use Match Rate and RMSE (Xie et al., 2022) to evaluate by matching the predicted candidates with the ground-truth structure (Ong et al., 2013). Match Rate is the proportion of the matched structures over the test set and RMSE is calculated between the ground truth and the best matching candidate, normalized by lattice volume and atom count (seeing Appendix F). We generate 1 sample and 20 samples of the same composition and consider a match if any sample aligns with the ground truth. For DNG task, we use Mean Absolute Error (MAE), RMSE and Tolerance Rate to evaluate how accurately our property-guided generation framework produces samples whose properties match the specified targets. Tolerance Rate measure the proportion of samples whose properties fall within a specified range of the target property.

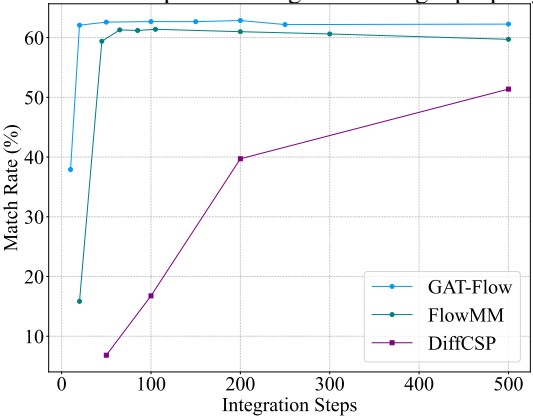

Figure 3: Match rate as a function of number of integration steps on MP-20. GAT-Flow achieves a higher maximum match with fewer integration steps.

## 5.2 CSP Evaluation Results

As shown in Table 1, our method not only achieves the highest match rate but also maintains a relatively low RMSE, demonstrating excellent prediction performance. Our method significantly outperforms other methods on complex dataset MPTS-52, highlighting the effectiveness of GAT network and the PC sampling strategy. Fig. 3 compares the match rates of DiffCSP, FlowMM and GAT-Flow as functions of integration steps to evaluate sampling efficiency. GAT-Flow achieves higher match rates with significantly fewer steps, indicating more efficient inference. Specifically, GAT-Flow reaches peak performance in about 20 steps, compared to FlowMM's 50 steps and DiffCSP's 1000 steps, reducing inference time by at least an order of magnitude.

We ablate each component of GAT-Flow in Table 2. Disabling the attention mechanism for either **F** or **L** leads to performance degradation, with match rates dropping to 62.25% and 61.57%, respectively, along with increased RMSE, demonstrating its importance in capturing fine-grained interactions between nodes. Moreover, replacing the PC sampling strategy with first-order Euler or second-order Taylor approximations further ODE solver reduces the match rate to around 60% and increases RMSE, highlighting the critical role of the PC strategy in ensuring generation stability and accuracy.

## 5.3 DNG Evaluation Results

In DNG scenarios where atom types are not fixed, an additional CNF is used to generate a one-hot encoded set of atomic types **A**. To maintain chemical relationships while reducing dimensionality, we employ the periodic atomic encoding method in (Luo et al., 2024) which is restructured into a 13x15 grid, with each element assigned a unique position (see Appendix E). In order to provide stable crystal structures for LLM in property-driven DNG task, we also evaluated the model's performance on the unconditional DNG task. Table 5 shows that GAT-Flow achieves well-balanced performance across various property metrics at substantially fewer integration steps compared to other large language models such as FlowLLM and CrystalLLM. These results confirm that our method is able to produce

Table 3: Results on property-driven DNG task. Statistics are computed on 10000 samples.Performance is evaluated using Mean Absolute Error (MAE), Root Mean Square Error (RMSE) and Tolerance Rate, with error thresholds set to 1eV, 0.5eV and 0.1eV, respectively. The best results are in bold.

| Method | MAE↓ | RMSE↓ | Tolerance Rate(%) ↑ | | |
|---|---|---|---|---|---|
| | | | $\varepsilon = 1eV$ | $\varepsilon = 0.5eV$ | $\varepsilon = 0.1eV$ |
| CrystalFlow (Luo et al., 2024) | 1.4453 | 1.7755 | 43.20 | 23.40 | 5.10 |
| MatterGen (Zeni et al., 2025) | 1.5204 | 2.0008 | 42.10 | 20.80 | 4.20 |
| GAT-Flow (ours) | **1.4136** | **1.7524** | **44.67** | **26.67** | **8.33** |

rational, novel and reliable crystal structures with fewer integration steps for property-driven DNG task.

Inspired by the exceptional capabilities of LLMs in multimodal understanding and generation tasks, we developed a text-guided framework for property-driven crystalline materials generation. Specifically, we fine-tuned an LLM to predict crystal properties from natural-language descriptions. During the crystal generation process, we then applied rejection sampling to discard candidates that failed to satisfy the target property thresholds, guiding the generation toward the desired properties.

In this work, we focus on the band-gap energy of materials. To target semiconductor candidates, we set the desired band-gap at 3.0 eV. Building on previous efforts Rubungo et al. (2023) and noting that T5 offers both faster inference and better prediction accuracy than BERT-based models, we chose T5 as our backbone. Following the LLM-prop setup, we added a single linear layer on top of the T5 encoder to perform band-gap prediction. For fine-tuning, we used the DFT-3D subset of the JARVIS-DFT database and selected those entries with TB-mBJ–computed band gaps, ensuring more precise prediction of semiconductor band-gap energy.

Leveraging the LLM's ability to predict band-gap energy from crystal descriptions, we steer the sampling process toward our target property via rejection sampling strategy. At each sampling step, we introduce both global and local perturbations to the crystal's graph features and use the fine-tuned LLM to predict the band-gap energy for each perturbed variant. We then select the structure whose predicted band-gap energy is closest to the target as the basis for the next iteration. To avoid mode collapse phenomena similar to those seen in GANs, this selection is performed only among perturbations derived from the same original sample.

For property-driven DNG task, we compared GAT-Flow with several representative methods Luo et al. (2024); Zeni et al. (2025), both of which adopt traditional conditional generation strategies based on diffusion models and flow models. In contrast, our proposed GAT-Flow employs an LLM-guided rejection sampling approach. As shown in Table 3, GAT-Flow outperforms the baselines across all evaluated metrics, demonstrating the effectiveness and feasibility of leveraging LLM guidance in property-driven DNG tasks.

## 6 CONCLUSION

The proposed framework, GAT-Flow, has emerged as a novel approach for crystalline materials discovery by integrating graph attention networks with flow-based modeling to effectively capture complex dependencies within crystalline systems. The incorporation of the Predictor-Corrector (PC) sampling strategy has further enhanced its capability to efficiently explore the vast structural and compositional space while preserving atomic-level accuracy.

Extensive experiments have demonstrated that GAT-Flow attains state-of-the-art performance in crystal structure prediction, underscoring its robustness and generalization ability. Additionally, the PC sampling strategy has been shown to improve prediction accuracy while reducing inference time. Furthermore, evaluation results indicate that incorporating a text-guided framework guided by a fine-tuned LLM enhances the model's ability to generate crystalline materials with specific properties. These findings underscore the potential of integrating flow-based generative models, graph-structured representations, and text-guided frameworks to advance computational materials discovery.

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

## A  CRYSTALS PRESENTATION AND SYMMETRIES

### A.1  EQUIVARIANCE AND INVARIANCE

Let $G$ be a group acting on spaces $\mathcal{X}$ and $\mathcal{Y}$. A function $f : \mathcal{X} \to \mathcal{Y}$ is said to be *G-equivariant* if it satisfies

$$\forall x \in \mathcal{X}, \ \forall g \in G, \quad f(g \cdot x) = g \cdot f(x),$$

and *G-invariant* if

$$\forall x \in \mathcal{X}, \ \forall g \in G, \quad f(g \cdot x) = f(x).$$

Since crystals cannot be uniquely defined by any specific representation $\mathbf{C}$, but rather by equivalence classes under symmetry transformations, we assume that the data distribution has a $G$-invariant density, where $G$ represents the symmetry group of the crystal.

## A.2 INVARIANT DENSITY.

If a function $f : \mathcal{X} \to \mathcal{Y}$ is $G$-equivariant and invertible, then the pushforward of a $G$-invariant density $p$ through $f$ remains $G$-invariant. That is, the statistical properties of crystals are preserved under symmetry transformations, which is essential for modeling and prediction tasks.

**Permutation Invariance**    The symmetric group $\mathfrak{S}_n$ acts by permuting the indices of atoms:

$$\sigma \cdot \mathbf{C} = ([\mathbf{A}^{\sigma(1)}, \ldots, \mathbf{A}^{\sigma(n)}], [\mathbf{F}^{\sigma(1)}, \ldots, \mathbf{F}^{\sigma(n)}], \mathbf{L}),$$

for all $\sigma \in \mathfrak{S}_n$. This reflects the indistinguishability of identical atoms within the crystal.

**Translation Invariance**    The translation group $\mathbb{T}^3$ acts by shifting all fractional coordinates modulo the unit torus:

$$\tau \cdot \mathbf{C} = (\mathbf{A}, \mathbf{F} + \tau\mathbf{1}^\top - \lfloor + \tau\mathbf{1}^\top \rfloor, \mathbf{L}),$$

where $\tau \in [-\frac{1}{2}, \frac{1}{2}]^3$, and $\lfloor \cdot \rfloor$ denotes element-wise floor operation. This ensures that translations do not alter the structure due to the periodic nature of crystals.

**Rotational Invariance**    The special orthogonal group $\mathrm{SO}(3)$ acts on the unit cell via rotations:

$$Q \cdot \mathbf{C} = (\mathbf{A}, \mathbf{F}, Q\mathbf{L}),$$

where $Q \in \mathrm{SO}(3)$. Note that the lattice parameters $\mathbf{L}$ encode geometric information without directional bias, hence the overall structure remains invariant under rotation.

## B    DETAILED BACKGROUND ON CSP AND DNG

### B.1    CRYSTAL STRUCTURE PREDICTION

In CSP task, the goal is to generate crystal structures conditioned on a fixed chemical composition. The formulation involves learning:

$$p(\mathbf{F}, \mathbf{L} \mid \mathbf{A}, \mathbf{c}),$$

where fractional coordinates $\mathbf{F} \in [0, 1)^{n \times 3}$, lattice parameters $\mathbf{L} \in \mathcal{L} \subset \mathbb{R}^6$, atomic types $\boldsymbol{a} \in \mathcal{A}^n$ and external conditions $\mathbf{c} \in \mathbb{R}^d$.

This setup ensures that generated structures strictly follow the specified chemical constraints, making it ideal for scenarios where the composition is known and fixed.

## B.2 DE NOVO GENERATION

For DNG task, the task is to explore new material compositions by learning a joint distribution:

$$p(\mathbf{F}, \mathbf{L}, \mathbf{A}) = p(\mathbf{F}, \mathbf{L} \mid \mathbf{A}) \cdot p(\mathbf{A}),$$

where $p(\mathbf{A})$ is a learned prior over atomic compositions.

To enable continuous modeling of discrete atom types, we map each atom type $\mathbf{A}_i \in \{1, \ldots, h\}$ to a binary vector $\boldsymbol{b}_i \in \{-1, 1\}^{\lceil \log_2 h \rceil}$. The full atomic configuration is then represented as:

$$\boldsymbol{b} = [\boldsymbol{b}_1; \ldots; \boldsymbol{b}_n] \in \{-1, 1\}^{n \cdot \lceil \log_2 h \rceil}.$$

At training time, a flow-based model transforms a standard normal distribution $\mathcal{N}(\mathbf{0}, \mathbf{I})$ into this binary latent space. During inference, the continuous output is discretized using the sign function:

$$\hat{\boldsymbol{b}} = \text{sign}(\mathbf{z}), \quad \text{where } \mathbf{z} \sim \mathcal{N}(\mathbf{0}, \mathbf{I}).$$

When $\lceil \log_2 h \rceil \neq \log_2 h$, some bit combinations do not correspond to valid atom types—these are "unused bits". The model effectively learns to ignore these during generation, ensuring chemically meaningful outputs.

### B.3 COMPARISON BETWEEN CSP AND DNG

While CSP task focuses on generating structures conditioned on fixed atomic types $a$, DNG task extends this framework by learning a distribution over $a$ itself. This enables the model to:

- Generate crystal structures with previously unseen atomic compositions,
- Explore the chemical space beyond known stoichiometries,
- Discover potentially novel materials with unique properties.

Together, CSP task and DNG task provide a comprehensive framework for crystal structure prediction and material design, covering both constrained and open-ended discovery settings.

## C  BACKGROUND ON CNF AND CFM

### C.1  CONTINUOUS NORMALIZING FLOWS

Continuous Normalizing Flows Chen et al. (2018) define a time-continuous transformation governed by an ordinary differential equation (ODE):

$$\frac{d\mathbf{z}(t)}{dt} = \mathbf{u}_\theta(\mathbf{z}(t), t),$$

where $\mathbf{u}_\theta$ is a neural network parameterizing the velocity field. Starting from a base distribution $p_0(\mathbf{z})$, typically standard normal $\mathcal{N}(0, I)$, the final state $\mathbf{z}(1)$ represents a sample from the learned distribution.

The change in log-density along the trajectory is given by:

$$\frac{d \log p_t(\mathbf{z}(t))}{dt} = -\mathrm{Tr}\left( \frac{\partial \mathbf{u}_\theta(\mathbf{z}(t), t)}{\partial \mathbf{z}(t)} \right).$$

This allows for exact density evaluation and gradient-based optimization, which is essential for modeling high-dimensional data like crystal structures.

## C.2 CONDITIONAL FLOW MATCHING

CFM introduces conditioning variables $\mathbf{y}$ into the generative process. The goal is to learn a function $g_\theta(\cdot; \mathbf{y})$ that maps noise $\mathbf{z} \sim p_0(\mathbf{z})$ to samples $\mathbf{x}$ distributed according to $p^*(\mathbf{x} \mid \mathbf{y})$.

The loss function used is:

$$\mathcal{L}_{\text{CFM}} = \mathbb{E}_{\mathbf{z}, \mathbf{y}} \left[ \|g_\theta(\mathbf{z}; \mathbf{y}) - \hat{\mathbf{x}}\|_2^2 \right].$$

To handle complex dependencies, CFM uses a conditional architecture that inputs both $\mathbf{z}$ and $\mathbf{y}$ jointly. In practice, we adopt the Independent Coupling variant (I-CFM) Liu et al. (2022); Albergo & Vanden-Eijnden (2022), where the vector field is conditioned on interpolation paths between initial and terminal points $(x_0, x_1)$. Specifically, the marginal path is defined as:

$$p_t(x) = \int p_t(x \mid z)q(z)dz,$$

and the corresponding vector field becomes:

$$u_t(x) = \int \frac{u_t(x \mid z)p_t(x \mid z)q(z)}{p_t(x)}dz.$$

Under I-CFM, the training objective simplifies to minimizing:

$$\mathcal{L}_{\text{CFM}}(\theta) = \mathbb{E}_{t, q(x_1, y), q(x_0)} \|v_{t;\theta}(t, x, y) - u_t(x \mid z)\|^2,$$

where $v_{t;\theta}$ is a time-dependent vector field modeled by a neural network.

**Proof of CNF training objective**  The CNF framework can be naturally extended to conditional generation with respect to a conditioning variable $y$. In this case, the evolution of the system is described by:

$$\frac{dx}{dt} = u_t(x \mid y), \tag{22}$$

$$p_t(x \mid y) = \int p_t(x \mid z)q(z \mid y)dz, \tag{23}$$

$$u_t(x \mid y) = \int \frac{u_t(x \mid z)p_t(x \mid z)q(z \mid y)}{p_t(x)}dz. \tag{24}$$

So the training objective is still $u_t(x \mid z)$.

# D  JOINT FLOW OF LATTICE AND COORDINATES

## D.1  MATRIX EXPONENTIAL AND INVARIANCE ANALYSIS OF LATTICE DEFORMATION

In our model, the lattice $L$ is represented through a rotation-invariant decomposition: $L = Q \exp\left(\sum_{i=1}^{6} k_i B_i\right)$, where $Q$ is a rotation matrix, $\exp(\cdot)$ denotes the matrix exponential function, and $k \in \mathbb{R}^6$ encodes deformation parameters. This representation ensures that regardless of how $k$ changes, $L$ maintains its rotational invariance, i.e., for any orthogonal transformation $O$, there exists an associated transformation matrix $O'$ such that $OL = LO'$.

## D.2 Derivation and Properties of the Minimum Image Convention

Considering periodic boundary conditions in crystal structures, we adopt the minimum image convention to calculate inter-atomic distances or displacement vectors. For any two position vectors $F_1, F_2$, their difference $\Delta F = F_1 - F_2$ can be adjusted as follows: $\Delta F' = w(\Delta F) = \Delta F - \lfloor \Delta F + 0.5 \rfloor$, ensuring that $\Delta F'$ lies within the primary cell. This operation is particularly useful for handling displacement vectors in fractional coordinate systems.

### D.3 FORM AND SAMPLING METHOD OF WRAPPED GAUSSIAN DISTRIBUTION

To maintain translational invariance under periodic conditions, we use the wrapped Gaussian distribution: $p_t^F(F \mid z) = \mathcal{N}_w(F; \mu_t^F(z), (\sigma^F)^2 I)$. Given the initial uniform distribution $\mathcal{U}([0,1)^{3N})$, the mean trajectory of the wrapped Gaussian is defined as: $\mu_t^F(z) = F_0 + t \cdot w(F_1 - F_0 - 0.5) + 0.5$, where $w(x)$ is as previously defined. This means that at each time $t$, we consider the shortest path within periodic boundary conditions.

## D.4 CONTINUITY OF VECTOR FIELD AND EXISTENCE AND UNIQUENESS OF ODE SOLUTIONS

For the proposed vector field $u_t(x \mid z) = (u_t^k, u_t^F)$, it is necessary to demonstrate that it satisfies the Lipschitz condition to guarantee the existence and uniqueness of solutions to the corresponding ordinary differential equations (ODEs). Specifically, for any $x, x' \in \mathbb{R}^{6+3N}$, $\|u_t(x \mid z) - u_t(x' \mid z)\| \leq L\|x - x'\|$, where $L$ is the Lipschitz constant. This property ensures that given an initial condition $(k_0, F_0)$, there exists a unique solution $(k_t, F_t)$ satisfying the defined dynamical equations.

## E    REORGANIZED PERIODIC TABLE OF ATOM TYPE FOR DNG

The elements of sub family, lanthanides and actinides are positioned in the bottom.

Table 4: Periodic Table of Elements

|    | 0  | 1  | 2  | 3  | 4  | 5  | 6  | 7  | 8  | 9  | 10 | 11 | 12 | 13 | 14 |
|----|----|----|----|----|----|----|----|----|----|----|----|----|----|----|----|
| 0  | H  | –  | –  | –  | –  | –  | –  | He | –  | –  | –  | –  | –  | –  | –  |
| 1  | Li | Be | B  | C  | N  | O  | F  | Ne | –  | –  | –  | –  | –  | –  | –  |
| 2  | Na | Mg | Al | Si | P  | S  | Cl | Ar | –  | –  | –  | –  | –  | –  | –  |
| 3  | K  | Ca | Ga | Ge | As | Se | Br | Kr | –  | –  | –  | –  | –  | –  | –  |
| 4  | Rb | Sr | In | Sn | Sb | Te | I  | Xe | –  | –  | –  | –  | –  | –  | –  |
| 5  | Cs | Ba | Tl | Pb | Bi | Po | At | Rn | –  | –  | –  | –  | –  | –  | –  |
| 6  | Fr | Ra | Nh | Fl | Mc | Lv | Ts | Og | –  | –  | –  | –  | –  | –  | –  |
| 7  | Sc | Ti | V  | Cr | Mn | Fe | Co | Ni | Cu | Zn | –  | –  | –  | –  | –  |
| 8  | Y  | Zr | Nb | Mo | Tc | Ru | Rh | Pd | Ag | Cd | –  | –  | –  | –  | –  |
| 9  | –  | Hf | Ta | W  | Re | Os | Ir | Pt | Au | Hg | –  | –  | –  | –  | –  |
| 10 | –  | Rf | Db | Sg | Bh | Hs | Mt | Ds | Rg | Cn | –  | –  | –  | –  | –  |
| 11 | La | Ce | Pr | Nd | Pm | Sm | Eu | Gd | Tb | Dy | Ho | Er | Tm | Yb | Lu |
| 12 | Ac | Th | Pa | U  | Np | Pu | Am | Cm | Bk | Cf | Es | Fm | Md | No | Lr |

# F    EVALUATION DETAILS

## F.1    EVALUATION DATA

Table 5: Results on unconditional DNG task. Statistics are computed on 10,000 samples. Performance is evaluated using structural and compositional validity, coverage recall and precision, and property statistics (density $\rho$; number of elements $N_{el}$), with the best results in bold.

| Method | Integ.Steps ↓ | Validity (%) ↑ | | Coverage (%) ↑ | | Property ↓ | |
| --- | --- | --- | --- | --- | --- | --- | --- |
| | | Structural | Compositional | Recall | Precision | wdist ($\rho$) | wdist ($N_{el}$) |
| CrystalLLM | - | 99.60 | **95.40** | 85.80 | 98.90 | 0.810 | 0.44 |
| FlowLLM | 250 | **99.94** | 90.84 | 96.95 | 99.82 | 1.140 | **0.150** |
| GAT-Flow (ours) | **100** | 99.65 | 81.75 | **98.95** | **99.84** | **0.324** | 0.255 |

## F.2 EVALUATION METRICS

Specifically, for each structure in the test set, we generate $k$ samples of identical composition and determine a match if at least one sample aligns with the ground truth, evaluated using the StructureMatcher class from pymatgen (Ong et al., 2013) with thresholds stol=0.5, angle_tol=10, and ltol=0.3. The *Match Rate* is defined as the ratio of matched structures to the total number of structures in the test set.

The Root Mean Square Error (RMSE) is computed between the ground truth and the best-matching candidate, normalized by $\sqrt[3]{V/N}$, where $V$ represents the lattice volume and $N$ is the number of atoms. This metric is averaged over all matched structures. For optimization methods, we select the 20 lowest-energy structures from the 5,000 generated during testing as candidates. In the case of generative baselines and our GAT-Flow model, we evaluate with $k = 1$ and $k = 20$.

In DNG task, we follow previous research works by employing the coverage metric to evaluate the structural and compositional similarity between the test set $\mathcal{S}_t$ and the generated structure set $\mathcal{S}_g$. Specifically, let $d_S(\mathcal{M}_1, \mathcal{M}_2)$ and $d_C(\mathcal{M}_1, \mathcal{M}_2)$ denote the L2 distances of the CrystalNN (Zimmermann & Jain, 2020) structural fingerprints and the normalized Magpie (Ward et al., 2016) compositional fingerprints, respectively. The Coverage Recall (Cov-R) is defined as:

$$\text{Cov-R} = \frac{1}{|\mathcal{S}_t|} \left| \{ \mathcal{M}_i \mid \mathcal{M}_i \in \mathcal{S}_t, \exists \mathcal{M}_j \in \mathcal{S}_g, d_S(\mathcal{M}_i, \mathcal{M}_j) < \delta_S, d_C(\mathcal{M}_i, \mathcal{M}_j) < \delta_C \} \right|, \quad (25)$$

where $\delta_S$ and $\delta_C$ are predefined thresholds. Similarly, the Coverage Precision (Cov-P) is defined by swapping $\mathcal{S}_t$ and $\mathcal{S}_g$. The recall metrics measure the number of correctly predicted ground-truth materials, while the precision metrics assess the quantity of high-quality generated materials.

### F.3 DATASET DETAILS

Our evaluation employs two distinct datasets, each presenting unique challenges. MP-20 is derived from a database of stable inorganic compounds, totaling 45,231 samples, each containing up to twenty atoms. MPTS-52 dataset offers a more complex scenario with 40,476 structures, each potentially containing up to fifty-two atoms, organized chronologically by their first publication year.

For the MP-20 datasets, we apply a standard split ratio of 60% for training, 20% for validation, and 20% for testing. Given its chronological nature, the MPTS-52 dataset uses a different splitting strategy: 27,380 samples for training, 5,000 for validation, and 8,096 for testing, ensuring that the temporal sequence of data is preserved throughout the experiments. This approach allows for a thorough assessment of our proposed methodologies across a wide range of material properties and complexities.

# G HYPERPARAMETERS TABLE FOR GAT-FLOW

Table 6: Training and test parameters of GAT-Flow on MP-20 and MPTS-52 datasets.

| | MP-20 | MPTS-52 |
|---|---|---|
| **Model** | | |
| Element type encoding dimension | 128 (28 in DNG task) | 128 |
| Gaussian expansion for pressure, number of bases | 80 | 80 |
| Gaussian expansion for pressure, bases start | $-2.0$ | $-2.0$ |
| Gaussian expansion for pressure, bases stop | 5.0 | 5.0 |
| Time sinusoidal positional encoding dimension | 256 | 256 |
| GAT hidden dimension | 512 | 512 |
| GAT number of layers $L$ | 8 | 7 |
| Number of frequency for $\mathbf{F}_{ij}^{\mathrm{FT}}$ | 256 | 256 |
| Loss weight $\lambda_k$ | 1 | 1 |
| Loss weight $\lambda_F$ | 10 | 10 |
| **Optimizer** | | |
| Optimizer type | Adam | Adam |
| Learning rate | $1e-3$ | $1e-3$ |
| Learning rate scheduler | ReduceLROnPlateau | ReduceLROnPlateau |
| Scheduler patience (epoch) | 40 | 40 |
| Scheduler factor (epoch) | 0.6 | 0.6 |
| Minimal learning rate | $1e-5$ | $1e-5$ |
| Training data batchsize | 256 | 64 |
| Training most epochs | 3000 | 3000 |
| **Inference** | | |
| Integration Steps | 200 | 100 |
| Anneal slope of coordinate | 7 | 5 |

All models are trained on RTX 3090. GAT-Flow for MP-20 dataset is trained on 4 RTX 3090 for 21.5 hours while GAT-Flow for MPTS-52 dataset is trained on 4 RTX 3090 for 74 hours.

# H USE OF LARGE LANGUAGE MODELS

We used a large language model to polish the writing of this manuscript (grammar, wording, and stylistic clarity) after the research, experiments, and analyses were completed. The model did not generate ideas, datasets, code, or empirical results. All content was reviewed and verified by the authors, who take full responsibility for the final text.

