# OpenReview forum: "GAT-Flow: Predictor-Corrector Flow Matching with Graph Attention Network for Crystalline Materials"
_ICLR.cc/2026/Conference — ICLR 2026 Conference Withdrawn Submission_

### Official Review · Reviewer_K4aX · 2025-10-27

**Soundness:** 1
**Presentation:** 1
**Contribution:** 1
**Rating:** 2
**Confidence:** 3

**Summary:**

This paper proposes a method for generation of materials using a flow-based model and making use of previous work on attention for graphs and corrector steps, and proposes a method for generating materials with certain properties, comparing with some previous work.

To me it seems that the contribution is a combination of previous works, and that this creates a SOTA method. However, I have some concerns about the evaluation and SOTA claims together with questions about the evaluation in general, and therefore do not think this paper is enough for acceptance. It should be noted that I don’t think SOTA is a prerequisite for acceptance if the method proposed has some other novelties and/or presents some new research direction/approach. However, I fail to see that the proposed method includes much technical novelty, and the empirical results are lacking. Additionally, I think the paper lacks clarity on the proposed method.

**Strengths:**

Works in a topic that has big promise. Finding synergies between previous works is useful research.

**Weaknesses:**

As the method relies on a combination of previous work, I think that it is important to see how this combination actually contributes to good performance. The authors seem to agree, as they stress their claimed SOTA results. However, I first of all have doubts about SOTA as they have not included DiffCSP++, which has better numbers (table 2 in the DiffCSP++ paper) on the CSP task (compared to numbers in first half of the table in the current paper, where a single sample has been generated). Secondly, I think it is weird that FlowMM has been omitted in the second half of Table 1. Finally, the numbers are not presented with any intervals over different seeds, and as GAT-Flow is extremely close in numbers compared to other methods, this to me raises doubts on how significant the improvements are.  Also related to the evaluation, the paper performs conditional DNG, but also provides some unconditional sampling. However, in this case they compare with two models based on LLMs, which I think is a bit strange choice. Why not any of the models used for the CSP task, and why specifically LLMs? As far as I understand, the proposed method only uses an LLM for **conditional** sampling.

Another weakness is that the paper is sometimes lacking details, making it difficult to understand what has been done, and impossible to reproduce the method. Some examples:
* Line 305: This section suddenly introduces some new notation, like $x_t$, potential function, Wiener process, etc. What does “which is a deterministic flow process that does not introduce new sample distribution models and is susceptible to pattern collapse” actually mean? Any references? Also, I am unsure what is meant by “corrector”. Typically, this word (in the context of predictor-corrector) means performing additional sampling steps (using the neural network) to try to correct the sample with respect to some distribution. However, I understand it like the “corrector” in this case is a mere addition of random noise?
* Line 341: how is $s(t)$ implemented in practice? I cannot find any details.
* Line 447: how is the fine-tuning of the LLM done?
* Line 459: what does “both global and local perturbations to the crystal’s graph features” mean?
* Line 462: “To avoid mode collapse phenomena similar to those seen in GANs, **this selection is performed only among perturbations derived from the same original sample**” I don’t understand what this means.

**Questions:**

Line 45: As part of the introduction, it is claimed that the lack of GAT in other works is a severe drawback which limits the possibility to model long-range interactions, leading to physically implausible results. Any references to back that up, or own experiments which show that models not based on GAT leads to **physically implausible results**, while your method (or other methods based on GAT) specifically avoids that?

Equation 10: should it be $\phi$ and not $\varphi$? (although these are technically the same letter, so maybe use some other notation for the Fourier transform)

Line 173 and 272: You refer twice to using the CFM framework, but provide different references. Why?

Line 377: what do you mean by “conditional” when referring to DiffCSP being a “**conditional** diffusion model”?

Table 3: The numbers are difficult to interpret. What is the spread of the errors (for example, a histogram would have been nice), and how does the unconditional model perform (to put the numbers in some perspective)?

What is the motivation for using an LLM and not some other classifier/regression method? How does the use of the LLM affect the computational costs, and how do you method compare with others from a computational standpoint (for example, how many “parallel” samples do you use in the rejection sampling, adding to the computational cost)?

---

### Official Review · Reviewer_dXJj · 2025-10-29

**Soundness:** 2
**Presentation:** 3
**Contribution:** 2
**Rating:** 4
**Confidence:** 4

**Summary:**

The paper introduces GAT-Flow, a novel flow-based generative model designed for crystal structure prediction (CSP) and de novo generation (DNG). The model incorporates a Geometric Graph Attention Network (GAT) to effectively capture both local and long-range atomic interactions. To enhance sampling stability and accuracy, it employs a Predictor–Corrector (PC) sampling strategy inspired by Langevin dynamics. Furthermore, GAT-Flow integrates a text-guided, property-driven generation module that leverages a fine-tuned T5 language model to enable the generation of materials with desired properties, such as specific band-gap values. The proposed framework achieves state-of-the-art performance on benchmark datasets including MP-20 and MPTS-52, and demonstrates strong results on various CSP tasks.

**Strengths:**

- The paper is very well written, with the authors effectively motivating the problem and providing sufficient background to establish the context. They have clearly articulated the key challenges, which enhances the overall understanding of the work.

- The three main contributions of the paper are: (1) the introduction of a geometric multi-layer Graph Attention Network (GAT) for modeling atomic interactions, (2) the development of a Predictor–Corrector sampling strategy to improve the stability and accuracy of the generation process, and (3) the integration of LLM-based text guidance for property-driven material generation. Additionally, the proposed framework demonstrates significantly improved speed and computational efficiency during the sampling process compared to existing methods.

- The proposed framework achieves state-of-the-art performance on benchmark datasets including MP-20 and MPTS-52, and demonstrates strong results on various CSP tasks.

**Weaknesses:**

- The authors claim that existing SOTA methods do not incorporate Graph Attention Networks (GATs), which limits their ability to capture long-range atomic interactions and may lead to physically implausible outputs. However, no empirical proof, examples, or supporting references are provided to substantiate this claim.

- CSP task results for important datasets such as Perov and Carbon are missing. Moreover, key baselines, including TGDMat[1], SymmCD[2], and CrysBFN[3] have not been compared.

- For the DNG task, the results are not very competitive. The authors compare with only a limited number of baselines, and both compositional and structural validity drop significantly. Compared to general diffusion-based models, the performance on property statistics is also lower. Authors should add more baselines like TGDMat[1] or CrysLLMGen[4]

 - The paper does not report key generation quality metrics such as stability, uniqueness, and novelty, which have become standard in recent material generation works.

- The property-conditioned generation experiments are limited to band-gap prediction only; other critical physical properties like formation energy and thermodynamic stability are not evaluated.

- Several important references and baselines are missing, including TGDMat[1], SymmCD[2], CrysLLMGen[4], CrysBFN[3], and UniMat[5].

- There are formatting issues in the appendix, with inconsistent spacing throughout.

- The source code is not available, making the results difficult to reproduce.

[1] Das, Kishalay, et al. "Periodic materials generation using text-guided joint diffusion model." ICLR 2025.

[2] Levy, Daniel, et al. "SymmCD: Symmetry-Preserving crystal generation with diffusion models." ICLR 2025.

[3] Wu, Hanlin, et al. "A periodic bayesian flow for material generation." ICLR 2025.

[4] S Khastagir, et al.   LLM Meets Diffusion: A Hybrid Framework for Crystal Material Generation. NeurIPS 2025.

[5] Yang, Sherry, et al. "Scalable diffusion for materials generation." arXiv preprint arXiv:2311.09235 (2023).

**Questions:**

Check the Weaknesses

---

### Official Review · Reviewer_iYNa · 2025-11-01

**Soundness:** 2
**Presentation:** 3
**Contribution:** 2
**Rating:** 2
**Confidence:** 3

**Summary:**

This paper proposes a novel flow-based generation approach GAT-Flow for crystalline materials. GAT-Flow adopts graph attention network as the backbone model and ODE based flow model as the generation framework. Experiments are conducted to show the promising performance of GAT-Flow.

**Strengths:**

- This paper proposes a novel method for crystalline material generation. The proposed approach of capturing periodic patterns with ODE-based flow framework is useful.
- Experiments show some promising performance of the proposed approach.
- The writing of this paper is generally good and excellent.

**Weaknesses:**

- A rigorous mathematic proof is needed to show why the proposed way (line 205 to 215) of applying ODE based flow on fractional coordinates satisfies translational invariance under periodic boundary.
- A major novelty contribution claimed by the paper is the usage of graph attention (GAT) network as the backbone model. However, there is not clear motivations, insights, or experimental studies about the advantages of using this architecture. How does GAT compare with other graph neural networks like GCN, GIN? If attention mechanism itself is important, why not using graph transformer architecture? Authors are strong encouraged to make more discussions and analysis about the choice of GAT so as to consolidate this novelty contribution claim.

**Questions:**

No additional questions.

---

### Official Review · Reviewer_FJno · 2025-11-02

**Soundness:** 2
**Presentation:** 4
**Contribution:** 2
**Rating:** 4
**Confidence:** 4

**Summary:**

This paper proposes GAT-Flow, a flow-based generative model that leverages Graph Attention Networks (GAT) for crystalline material generation and prediction. It jointly predicts lattice vectors and atomic coordinates while preserving crystal symmetries using a geometric multi-layer attention mechanism. A novel Predictor–Corrector (PC) sampling strategy enhances efficiency and numerical stability during generation. Additionally, a language model-guided framework enables property-driven material design based on text prompts. Experimental results on benchmark datasets show state-of-the-art performance in both crystal structure prediction and de novo material generation with desired properties.

**Strengths:**

This paper proposes GAT-Flow, a flow-based generative model that leverages Graph Attention Networks (GAT) for crystalline material generation and prediction. It jointly predicts lattice vectors and atomic coordinates while preserving crystal symmetries using a geometric multi-layer attention mechanism. A novel Predictor–Corrector (PC) sampling strategy enhances efficiency and numerical stability during generation. Additionally, a language model-guided framework enables property-driven material design based on text prompts. Experimental results on benchmark datasets show state-of-the-art performance in both crystal structure prediction and de novo material generation with desired properties.

**Weaknesses:**

1. Authors claims SOTA methods do not incorporate Graph Attention Network (GAT), which limit their ability to capture long-range interactions, leading to physically implausible outputs. Any proof on that? Any supporting examples or reference would be great to understand these claim.

3. CSP task results for Perov and Carbon are missing. Also baselines like TGDMat, SymmCD, CrysBFN are not there.

2. Results for DNG task is not good. Firstly very limited baselines are compared. Moreover compositional validity goes down significantly and structural too decreases. Compared to general diffusion models performance is lower for property stat.

3. Authors didnot report stability, uniqueness and novelty results are which are recently considered as primary metric for quality of material generation.

4. Only band-gap prediction is used for property conditioned generation; other physical properties (formation energy, stability) aren’t tested.

4. Source Code not available. Cant reproduce.

**Questions:**

See weakness

---

### Note · Authors · 2025-11-26

**Comment:**

**Subject:** Withdrawal Request:15095

**Dear Reviewers,**

We would like to express our sincere gratitude to you and the reviewers for the time and effort dedicated to evaluating our work. The constructive feedback and detailed comments we received have been incredibly valuable to us.

After careful consideration of the reviewers' suggestions, we realize that the manuscript requires further refinement and additional work to fully address the points raised. Therefore, we have decided to withdraw the paper at this stage to focus on incorporating these insights and thoroughly polishing the manuscript.

We apologize for any inconvenience this withdrawal may cause and thank you for your understanding and support.

**Withdrawal Confirmation:**

I have read and agree with the venue's withdrawal policy on behalf of myself and my co-authors.